# Differences in the Time Course of Recovery from Brain and Liver Dysfunction in Conventional Long-Term Treatment of Wilson Disease

**DOI:** 10.3390/jcm12144861

**Published:** 2023-07-24

**Authors:** Harald Hefter, Theodor S. Kruschel, Max Novak, Dietmar Rosenthal, Tom Luedde, Sven G. Meuth, Philipp Albrecht, Christian J. Hartmann, Sara Samadzadeh

**Affiliations:** 1Departments of Neurology, University of Düsseldorf, Moorenstrasse 5, D-40225 Düsseldorf, Germany; theo.kruschel@gmx.de (T.S.K.); max.novak@uni-duesseldorf.de (M.N.); dietmar.rosenthal@med.uni-duesseldorf.de (D.R.); svenguenther.meuth@med.uni-duesseldorf.de (S.G.M.); philipp.albrecht@med.uni-duesseldorf.de (P.A.); christian.hartmann@med.uni-duesseldorf.de (C.J.H.); sara.samadzadeh@yahoo.com (S.S.); 2Departments of Gastroenterology, University of Düsseldorf, Moorenstrasse 5, D-40225 Düsseldorf, Germany; tom.luedde@med.uni-duesseldorf.de; 3Department of Neurology, Kliniken Maria Hilf GmbH Mönchengladbach, 41063 Mönchengladbach, Germany; 4Charité–Universitätsmedizin Berlin, Corporate Member of Freie Universität Berlin and Humboldt-Unverstät zu Berlin, Experimental and Clinical Research Center, 13125 Berlin, Germany; 5Department of Regional Health Research and Molecular Medicine, University of Southern Denmark, 5230 Odense, Denmark; 6Department of Neurology, Slagelse Hospital, 4200 Slagelse, Denmark

**Keywords:** Wilson disease, spectrum of symptoms, recovery, cholinesterase, biomarker, orthotopic liver transplantation

## Abstract

Background: The aim of this study was to demonstrate that both neurological and hepatic symptoms respond to copper chelation therapy in Wilson disease (WD). However, the time course of their recovery is different. Methods: Eighteen patients with neurological WD from a single specialized center who had been listed for liver transplantation during the last ten years and two newly diagnosed homozygous twins were recruited for this retrospective study. The mean duration of conventional treatment was 7.3 years (range: 0.25 to 36.2 years). A custom Wilson disease score with seven motor items, three non-motor items, and 33 biochemical parameters of the blood and urine, as well as the MELD score, was determined at various checkup visits during treatment. These data were extracted from the charts of the patients. Results: Treatment was initiated with severity-dependent doses (≥900 mg) of D-penicillamine (DPA) or triethylene-tetramin-dihydrochloride (TRIEN). The motor score improved in 10 and remained constant in 8 patients. Worsening of neurological symptoms was observed only in two patients who developed comorbidities (myasthenia gravis or hemispheric stroke). The neurological symptoms continuously improved over the years until the majority of patients became only mildly affected. In contrast to this slow recovery of the neurological symptoms, the MELD score and liver enzymes had already started to improve after 1 month and rapidly improved over the next 6 months in 19 patients. The cholinesterase levels continued to increase significantly (*p* < 0.0074) even further. One patient whose MELD score indicated further progression of liver disease received an orthotopic liver transplantation 3 months after the diagnosis of WD and the onset of DPA treatment. Conclusions: Neurological and hepatic symptoms both respond to copper chelation therapy. For patients with acute liver failure, the first 4 months are critical. This is the time span in which patients have to wait either for a donor organ or until significant improvement has occurred under conventional therapy. For patients with severe neurological symptoms, it is important that they are treated with fairly high doses over several years.

## 1. Introduction

Wilson disease (WD) is a recessively inherited disorder of copper metabolism predominantly affecting the liver and brain that was named after S.A.K. Wilson [1]. Inspecting post-mortem WD brains, he was impressed by the damage to the brain, especially to the putamen (the lenticular nucleus), in addition to liver cirrhosis. Therefore, he called this disease entity progressive hepatolenticular degeneration [1].

About 110 years later, knowledge about the pathophysiology of WD has considerably improved [2,3]. The essential trace element copper, which is necessary for iron oxidation in the mitochondria and a variety of other enzymatic intracellular reactions, turned out to play a crucial role in this disease [3,4,5]. Copper is taken up from the gut and the portal veins by the human copper transporter (CTR1/hCTR1 [6]) into hepatocytes and transported to the trans-Golgi network. Here, the P-adenotriphosphate protease ATP7B (ATP7B) modifies apo-ceruloplasmin to ceruloplasmin (CER) by incorporating copper [7]. Intact ceruloplasmin regulates iron metabolism and the transport of copper to non-hepatic cells [7,8]. Furthermore, ATP7B is necessary for the excretion of excessive copper into the bile [9,10].

In 1993, causal mutations in the *ATP7B* gene (locus 13q14.3-q21.1) that are responsible for the development of WD were identified [11]. During the last 30 years, more than 1000 mutations have been reported [12,13]. However, there seems to be no genotype–phenotype correlation [14]. Even in homozygotic WD twins, the phenotype may be considerably discordant [15,16,17].

The spectrum of both hepatic and neurological symptoms is broad. On the one hand, WD may become manifest as fulminant liver failure [18]; on the other hand, it may be diagnosed in elderly people without or with mild hepatic symptoms by chance [19]. Neurologic WD may become manifest as a severe generalized movement disorder [2,3,15] or a mild tremor of the hands, voice, and tongue only [20]. These differences in its clinical manifestation are still poorly understood. Since neurological manifestations usually occur later than hepatic manifestations, the blood–brain barrier (BBB) is thought to protect against brain damage due to the influx of copper [21]. However, even that is doubted since free copper can penetrate and damage the BBB [22,23].

The response to treatment is similarly as broad as the clinical manifestations. Different symptoms respond differently to therapy [24]. Tremors seem to improve faster and better than dystonia in WD [24]. Even in homozygotic twins, the response to therapy, including liver transplantation, and the spectrum of the side effects of WD-specific therapy may be different [15,25]. However, the response to long-term treatment was not the only difference for different symptoms; additionally, the speed of improvement is different for different symptoms in WD.

To demonstrate this clearly and provide a solid base of information for advising newly diagnosed patients on what can be expected realistically during continuous WD-specific treatment, the time course of improvement in neurological and hepatic symptoms was compared in 20 WD patients who had been listed for orthotopic liver transplantation (LTX) but underwent conventional treatment.

## 2. Materials and Methods

This retrospective study was performed according to the Declaration of Helsinki and approved by the local ethics committee of the University of Düsseldorf (Germany).

### 2.1. Patient Recruitment

In the Clinic of Neurology of the University Hospital in Düsseldorf (Germany), a special outpatient department ward for rare metabolic diseases was implemented in 1985. About 1 to 3 new WD patients per year have been diagnosed in this institution since then. For the present retrospective study, 20 new WD patients were recruited. Seventeen fulfilled the following criteria: (i) their WD had been diagnosed in our department, (ii) the patient had been listed for liver transplantation (LTX) by colleagues from the regional departments of gastroenterology, and (iii) the patient had a well-documented course of conservative treatment since the onset of therapy in our institution. In addition, the data included one girl who (i) was diagnosed in a nearby city, (ii) was listed for liver transplantation, and (iii) whose mother had documented the course of treatment from the very beginning. Furthermore, two newly diagnosed homozygous twins with different disease severity and symptoms were included; their clinical data have already been presented elsewhere [15]. Although there was an indication for LTX in one of the twins, both had not been listed for LTX since both patients did not want to be transplanted. In summary, apart from the twins, all other patients had been listed for LTX.

The diagnosis of WD was based on an analysis of the Kayser–Fleischer rings, anterior segmental optical coherence tomography [26], cranial magnetic resonance imaging, acoustic radiation force impulse investigation [27,28], and typical biochemical findings.

### 2.2. Scoring of the Neurological Findings

At each therapeutic checkup visit in our outpatient department, the WD patients underwent a detailed neurological examination and were asked for their actual body weight. Then, the customized Düsseldorf Wilson Disease score (DWDS) was determined. This scale scores whether 10 specific symptoms are mildly (1 point), moderately (2 points), or severely (3 points) present or absent (0 points). The seven motor items of the DWDS (dysarthria/dysphagia, dystonia, bradykinesia, tremor, gait, cerebellar symptoms, and ophthalmological/brain stem symptoms) were summed up to yield the motor score (MotS; range: 0–21 points), and the three non-motor items of the DWDS (reflexes, sensory symptoms, and neuropsychiatric symptoms) were summed to yield the non-motor score (N-MotS; range: 0–9 points). MotS and N-MotS were summed to yield a total score (TS; range: 0–30 points). This scoring system can be completed by a treating physician within 1 min after a neurological examination and has been used in our institution since 1985 [29]; it is similar to a score used by the Italian OLT study group [30] and covers most of the neurological findings described by Shribman et al. [31].

### 2.3. Analysis of the Biochemical Parameters

After the neurological examination, a blood sample was taken. Furthermore, the WD patients in our department have been trained to collect their urine over 24 h without medication after previous cessation of WD-specific medication for 2 days and to bring a sample of the 24 h urine collection with them. For monitoring the therapy, 33 biochemical parameters were determined from the blood or the urine sample. These 33 biochemical parameters included: (i) four parameters of copper metabolism (ceruloplasmin (CER), serum copper (CUS), copper concentration in the 24 h urine collection (24 h-UCU/L), copper excreted in the urine over 24 h (24 h-UCU/d); (ii) four liver enzymes (AP, AST, ALT, and GGT) and pseudocholinesterase (CHE); (iii) four parameters testing kidney function (serum level of creatinine (CREA), clearance of creatinine in the 24 h urine collection (24 h-CREAC), concentration of protein excreted over 24 h (24 h-PROT/L), total amount of protein excreted over 24 h (24 h-PROT/d)); (iv) five parameters of the coagulation system (thrombocyte count (THROM), thromboplastin time (PTT), Quick´s test (Quick), international normalized ratio (INR), serum level of fibrinogen (FIBR)); and (v) four parameters of the iron metabolism (serum levels of iron (FE), transferrin (TRANS), ferritin (FERR), and hemoglobin (Hb), and erythrocyte count (ERY)). The other parameters were albumin (ALB), the serum level of bilirubin (BILI), leucocyte counts (LEUCC), and nine more parameters. As a further parameter, the MELD score was determined (MELD score = 10 × (0.957 × ln(CREA) + 0.378 × ln(BILI) + 1.12 × ln(INR) + 0.643)). These 34 parameters were determined at each checkup visit by the central laboratory of the University Hospital of Düsseldorf (Germany).

Demographic- and treatment-related data, body weight, the DWDS score, and biochemical findings were extracted from the charts of the patients.

### 2.4. Statistics

Patients were subdivided into patients with a MELD score of ≤10 and patients with a MELD score of >10. The parameters determined at the first visit to our department (initial data) were compared with the parameters determined at enrollment in this study (final data). The significance level was set to *p* = 0.05. Bonferroni’s alpha adjustments were applied for multiple comparisons. A two-group repeated measurement ANOVA was calculated to detect significant differences between groups and among repeated measurements for all 34 parameters of the blood and urine. For the correlation analysis, the rank correlation coefficient was determined. When a regression line was calculated, Pearson’s correlation coefficient (PCC = r) was also determined. ANOVA, rank correlation, and PCC were conducted/calculated using the commercially available SPSS statistics package (version 25: IBM Analytics, Armonk, NY, USA).

## 3. Results

### 3.1. Demographic and Treatment-Related Data and Spectrum of Initial Clinical Symptoms

The age at recruitment varied between 23.3 and 54.7 years (median: 38.2 years; interquartile range (Q25–Q75): 21 years). The age at the manifestation of WD varied between 7.5 and 50.4 years (median: 19.5 years; interquartile range: 12.1 years). Twelve patients were females; eight were males.

In Table 1, the clinical type of WD (asymptomatic, hepatic, etc.) is presented, as well as the result of the initial neurological investigation. Four patients did not have any neurological symptoms (TS = 0). Seven (35%) patients were classified as being mildly affected (total score: TS < 3), seven patients were moderately affected (2 < TS < 7), and six (30%) were severely affected (TS > 6). The initial TS did not correlate with age at manifestation of WD.

For the initial copper elimination treatment (CET), D-penicillamine (DPA) was used in 13 (65%) patients, and trientine dihydrochloride (TRIEN) was used in 6 (30%) patients. One asymptomatic patient remained on zinc monotherapy until he developed a hand tremor. He then agreed to be treated with TRIEN (for details, see [25]). The mean dose of DPA was 1131 mg (SD = 427), and the mean TRIEN dose was 1050 mg (SD = 266).

Retrospective analysis showed that, apart from two exceptions (the open circles in Figure 1), the dose of DPA chosen for the initial treatment was significantly correlated (r = 0.7545, *p* < 0.01) with the severity of WD (TS; full circles). Apart from two further cases (the open triangles in Figure 1), the dose of TRIEN was also correlated with TS, but because of the lower number of patients, this correlation was not significant (r = 0.833, n.s.) (Figure 1; full triangles). Body weight did not correlate with either the dose of DPA (r = 0.440, n.s.) or of TRIEN (r = 0.297, n.s.).

### 3.2. Course of Treatment and Improvement in Neurological Symptoms

During the course of treatment, the dose of DPA was further increased in three patients, remained constant in five patients, and reduced from 900 to 600 mg in one patient. In one patient, the treatment was switched from 900 mg TRIEN to 900 mg DPA. After a duration of 10.6 years (SD: 13.3), the mean dose of DPA was 1309 mg (SD: 345). One patient received DPA until he underwent LTX. The other 10 patients received TRIEN. After a duration of 17.4 years (SD: 10.4), the mean dose of TRIEN was increased to 1325 mg (SD: 381). Neither the increase in the dose of DPA nor the increase in the dose of TRIEN was significant. At recruitment, neither the dose of DPA nor the dose of TRIEN was significantly correlated with TS or body weight.

In 10 patients, MotS improved, whereas in 8 other patients, MotS remained constant. In only two patients, MotS worsened. These two patients developed comorbidities: one female patient (Patient 8) developed myasthenia gravis, and one male patient (Patient 19) had a left-hemispheric stroke (hatched arrow in Figure 2A,B). When TS was plotted against time since the onset of therapy, the severity of WD (TS) approached a mildly affected level (TS < 3) in the majority of patients (Figure 2A: dark grey area). In some patients, however, TS remained in the moderately affected range (2 < TS < 6; light grey area in Figure 2A). When TS was plotted against age at investigation, it was obvious that all patients aged below 37 years showed an improvement (Figure 2B). Some patients above the age of 37 (indicated by the vertical hatched line in Figure 2B) experienced a worsening of their condition, which was either due to comorbidities or due to a worsening of N-MotS. TS did not correlate with age at recruitment.

### 3.3. Improvement in the MELD Score and Biochemical Parameters

At the first presentation, 10 patients had a MELD score of >10. Within fewer than 240 days of treatment, the MELD score was less than 12 in all patients (Figure 3A), except in Patient 3, who received a transplant after 3 months of conservative treatment and whose data are presented in more detail below. The copper concentration in the 24 h urine collection showed high variability and decreased down to values below 0.030 mg/L in most of the patients during the first 700 days of treatment (Figure 3B). Apart from one exception, pseudocholinesterase (CHE) showed a continuous increase during a period of treatment of 700 days (Figure 4A). The serum levels of liver enzymes rapidly improved during the first 200 days of treatment and approached normal levels. This was also demonstrated for the levels of AST in Figure 4B.

In Table 2 (upper part), seven parameters for which a significant change could be detected under CET by rm-ANOVA are presented in detail. The two parameters with the most significant change after the onset of CET were the concentration of protein excreted (24 h-PROT/L) and the total amount of protein excreted daily (24 h-PROT/d). This implies that kidney function has to be carefully controlled during copper chelation therapy. Improvements in cirrhosis of the liver were demonstrated by the significant increase in CHE and the significant decrease in AP and AST levels. The significant increase in albumin (ALB) indicated significant improvements in protein synthesis. A variety of parameters did not show significant changes under CET because of the large initial interindividual variability.

When patients were split into 10 patients with an initial MELD score of >10 and 10 patients with an initial MELD score of ≤10, nine parameters showed a highly significant (*p* < 0.01) difference between these two groups (Table 2 (middle part)) and seven other parameters had a significant (*p* < 0.05) difference (not presented). Only three parameters (PTT, ceruloplasmin, and transferrin) remained significantly (*p* < 0.05) different under CET (Table 2 (lower part)). This underlines the excellent recovery of the biochemical parameters, especially in the patients who were initially more affected.

### 3.4. Liver Transplantation in a Wilson Disease Patient with Acute Liver Failure

A 19-year-old male patient (Patient 3) noticed fatigue and reduced mental drive and capacity in school. Laboratory testing detected elevated liver enzymes, increased serum levels of bilirubin, and an elevated INR, resulting in a MELD score of 22. Acute liver failure was diagnosed. Extensive testing for liver infection or autoimmune hepatitis was negative. He was admitted to our institution for further examination. Detailed laboratory testing revealed highly elevated copper excretion in the urine over 24 h and a decreased serum level of ceruloplasmin. Therefore, WD was diagnosed. The neurological examination and a slit-lamp investigation of the cornea were normal, as was the cranial MRI scan.

The patient was taken to a gastrointestinal ward, and DPA therapy was initiated. From the very beginning, a debate arose among the treating physicians about whether this patient should undergo LTX or whether he could be kept on conservative therapy. On the one hand, it was argued that the patient had a progressive increase in his MELD score, indicating a higher risk for mortality [32]. On the other hand, the patient was positively genetically tested for the presence of Gilbert syndrome (Morbus Meulengracht). This implied that the MELD score overestimated the degree of liver disease. In Figure 5A, the MELD score of the patient (full circles) is presented in comparison with a corrected MELD score (MELD score (corr), full diamonds) calculated under the assumption of a normal serum level of bilirubin. Furthermore, the cholinesterase started to increase (Figure 5B), and the liver enzymes (Figure 5C: AST (full circles) and ALT (full diamonds)) decreased 40 days after the onset of CET. Intermittently, the patient received antibiotic treatment several times, which led to transient increases in his liver enzymes (Figure 5C).

When a suitable young donor organ was allocated to the center, the patient successfully received a transplant without any perioperative complications about 4 months after onset of CET. His fatigue and mood disturbances rapidly improved, and his parameters of copper metabolism normalized within 3 months, and the patient went back to school again. Another extensive test for the presence of viral infections demonstrated that the patient had become EBV-positive after the transplantation.

### 3.5. Recovery of Pseudocholinesterase in 10 Selected Patients

In this section, the patient receiving the transplant (full circles in Figure 6B–D) is compared with seven new WD patients with subacute liver failure (open circles in Figure 6B–D) who had been listed as candidates for LTX but had conservatively been treated with copper chelation therapy, and two new conservatively treated WD patients who were homozygous twins (open squares in Figure 6B–D).

In the seven conservatively treated WD patients, excellent recovery of liver function was observed. This is demonstrated in Figure 6A, where the initial values of CHE (open circles) are plotted against the age at the onset of WD, as well as the best values of CHE (grey dots) under CET. The non-linear regression analysis between the best value of CHE and age revealed a highly significant (r = 0.9776; *p* < 0.001) age-dependent recovery from liver disease.

In Figure 6B, the temporal development of CHE during the first 125 days of treatment is presented for all 10 patients. The value on Day 125 was interpolated for all seven conservatively treated patients (open circles), the homozygous twins (open squares), and then extrapolated in the patient receiving the (full dots) since he underwent transplantation before Day 125. In the patient with the transplant and in the twins, a further small initial decline in CHE after initiation of CET can be seen (Figure 6B). After about 40 days, CHE started to improve. The mean initial CHE of the seven conservatively treated patients was 1541 U/L (SD: 349); after 125 days of treatment, CHE significantly (*p* < 0.043) increased to a mean of 2218 U/L (SD: 598). The increase in CHE during the first 125 days was age-dependent (r = −0.549; *p* < 0.05 (one-sided testing)). In Figure 6D, the temporal development of the MELD score is presented for all 10 patients during the first 125 days of CET. In four patients, a clear improvement (>2 score points) was seen in the MELD score, but in five patients, only a small change (−1 to +2) could be observed. Later on, in the patient who received the transplant, the MELD score clearly worsened during conservative treatment.

## 4. Discussion

### 4.1. The Broad Spectrum of Neurological Symptoms in Wilson´s Disease before and after Therapy

The spectrum of clinical neurological symptoms is broad in Wilson disease [2,3,21]. A variety of factors influence the phenotype [15]. Differences in copper exposure, differences in hormonal status including pregnancy, possible modifier genes, and differences in genotype and in the function of the blood–brain barrier may lead to differences in the clinical manifestation of WD [15]. To score the main neurological symptoms, we used a simple score (DWDS; see the Methods), which can be easily completed within 1 min after a clinical neurological investigation. It covers most of the symptoms mentioned in previous reports on the neurological symptoms of WD [3,31,33]. In the present cohort, only one patient had involuntary choreatic movements (Patient 10), which were not scored by the DWDS. In a much larger sample of 115 WD patients, only two patients had chorea and had not been tested for additional comorbidity with benign hereditary chorea or Huntington´s disease [33].

In general, the neurological symptoms manifest after the hepatic symptoms [21], but in the present cohort, no significant positive correlation between age and the severity of neurological symptoms could be detected. Retrospective analysis revealed that apart from four exceptions, the dose of CET was significantly correlated with the severity of the initial symptoms (Figure 1). This correlation disappeared during the course of treatment simply because the dose was kept constant or slightly increased, whereas the symptoms improved, especially in the more affected patients.

The spectrum of symptoms remained broad, but the frequency of individual symptoms changed since the sensitivity to CET is different for different symptoms, as described previously [24]. Motor symptoms improved in 50% of the patients and remained constant in a further 40% of the patients. Tremors and cerebellar symptoms responded best to CET, as reported previously, whereas bradykinesia or dystonia did not show many changes during therapy [24,29].

The present study shows clearly that the neurological symptoms improved over time. However, continuous treatment with fairly high doses (≥900 mg DPA or TRIEN) over several years seemed to be necessary for the majority of WD patients until they became only mildly affected.

In a cross-sectional study, the motor symptoms of WD patients under long-term treatment did not significantly change with age, in contrast to the non-motor symptoms [34]. This is in line with the present longitudinal observations (Figure 2), which showed an improvement in all patients aged <37 years and a mild worsening in some patients older than 37 years. Initial improvements and a secondary worsening for various reasons beyond the age of 40 have also been observed in a much larger cohort of 115 WD patients under long-term treatment [33].

### 4.2. Improvement in Biochemical Parameters during the Course of Therapy in Wilson´s Disease

The two-group rm-ANOVA revealed a significant time × group interaction: many more parameters improved during CET in the patients with a MELD score of >10 than in the patients with a MELD score of ≤10. Impaired coagulation and bleeding are critical complications of acute liver failure [18,35]. INR and PTT increased significantly, and thrombocyte counts were significantly reduced, indicating that, initially, different components of the coagulation system were affected in the patients with a MELD score of >10. Impaired coagulation was the main reason why Patient 3 received a transplant (see Section 3.4). Significantly elevated levels of liver enzymes (AP, AST, ALT, GGT) and decreased levels of CHE indicated liver impairment and the beginning of cirrhosis, whereas elevated levels of ferritin and decreased levels of transferrin and hemoglobin indicated an impairment in iron metabolism in the untreated WD patients with a MELD score of >10. However, all these parameters responded excellently to CET. Only PTT and transferrin levels remained different between the two groups of patients under CET. We believe that the use of rather high doses of DPA or TRIEN to reduce neurological symptoms was the reason why many biochemical parameters responded so well and quickly.

### 4.3. Difference in Time Course of the Improvement in Neurological and Hepatic Symptoms

In the brains of WD patients post-mortem, the copper content is elevated [36]. Over the course of therapy, the brain’s copper content decreases [34]. In WD patients, the brain’s metabolism recovers over the duration of therapy [37]. However, doses of copper chelators below 900 mg do not seem to be high enough to maintain the initial level of improvement in neurological symptoms reached during continuous treatment during the first years after the diagnosis of WD [37]. A low serum level of free copper seems to be necessary to guarantee the continuous efflux of copper from the brain of a WD patient, especially from structures with a high affinity to copper, such as the basal ganglia [38]. Therefore, continuous treatment with rather high doses seems to be necessary to reduce the neurological symptoms slowly over the years (Figure 2A).

The blood–brain barrier (BBB) probably plays a crucial role in the influx and efflux of copper to and from the brain [23,39]. Copper is an ion that has to be actively transported across the BBB. Thus, the BBB also reduces the efflux of copper from the brain into the blood under CET. The initial worsening of neurological symptoms after the onset of DPA therapy has been explained by the mobilization of free copper in combination with DPA-induced damage to the BBB and a subsequent increase in the influx of copper to the brain [23,39]. With a fast initial increase in the dose of DPA or TRIEN over 3 weeks, we did not observe this initial worsening and therefore think that the efflux of copper from different structures of the brain with different levels of affinity to copper is also a plausible explanation for the initial worsening. Whether neurological symptoms respond better to tetramolybdate (which does not damage the BBB [23]) than to DPA has yet to be analyzed in long-term cross-over studies.

It has been observed previously that the neurological symptoms in WD may recover greatly within months after LTX [40,41]. This has recently been confirmed again [42]. Nevertheless, it seems to be a challenging task to convince transplantation centers to operate on WD patients because of the neurological indications.

In contrast to the slow improvement in neurological symptoms over years under conventional treatment, the biochemical parameters of the blood and urine responded rapidly within weeks after the onset of CET. A decline in the copper concentration in the 24 h urine collection looked like a wash-out curve (see Figure 3B). In parallel, the liver enzymes improved, and the CHE recovered continuously (Figure 4A,B). In most of the patients, the serum levels of liver enzymes approached the normal range within 200 days. This recovery seemed to depend on the age of the patient (see Figure 6). However, this recovery also depends heavily on the compliance of the patient and the velocity of the increase in the initial dose.

### 4.4. Is There a Time Window for Conservative Therapy in Patients with Acute Liver Failure?

The present study, on the one hand, confirmed that the CHE level is a sensitive biomarker for detecting untreated WD [43]. On the other hand, it indicated that CHE measurements may also be an appropriate tool for monitoring the conventional treatment of WD. The analysis of the levels of liver enzymes and CHE in patients with an initial CHE of <2000 U/L revealed that an improvement in liver function can be expected from Days 30 to 40 after the initiation of CET (Figure 6). After 125 days of therapy, the recovery from liver dysfunction had clearly progressed, and after 200 days of continuous CET, the danger arising from liver failure [18,35] seemed to be over. Since the time from listing for LTX to the availability of a suitable donor organ usually lasts several months in an industrialized country [44,45], there is a realistic chance that LTX may not be necessary after 200 days as long as conservative therapy has been performed consequentially for several months (see also [46]).

Nevertheless, the example of Patient 3, who received a transplant, clearly demonstrated that the MELD score may continue to increase although the liver enzymes have started to improve (see Figure 5). This is a clear sign of a life-threatening situation [32] and a clear indication of LTX.

There is hope that, in the future, even more rapidly acting copper-eliminating substances than DPA or TRIEN will become available. Methanobactin, a yeast product, has an extremely high affinity for copper [47,48]. In a rat model of WD (LEG-rat), methanobactin improved mitochondrial dysfunction within days after the onset of treatment and reversed acute liver failure [49,50]. There is good reason to assume that this will also happen in humans; however, applications of methanobactin in humans are lacking so far.

## 5. Conclusions

In WD, both the neurological and hepatic symptoms respond to copper elimination therapy (CET) quite well. However, the hepatic symptoms respond much faster than the neurological symptoms. Within 200 days of treatment, the MELD score declined to values around 10, and liver enzyme levels returned to normal values, whereas some neurological symptoms may persist over several years despite continuous treatment. CET should be initiated with sufficiently high doses of DPA or TRIEN to reduce the neurological impairment to a mild level, which would allow a fairly normal life. The patients in the present study were treated with rather high doses. A subsequent study is recommended to prospectively analyze whether patients with mild or no neurological symptoms should be treated with doses as high as those given to patients with moderate or severe neurological findings. In the case of acute liver failure, conservative treatment with doses above 900 mg of DPA or 1200 mg of TRIEN during the first 4 months after the diagnosis of WD may be sufficient to improve liver function to such an extent that LTX is not required.

## 6. Strengths and Limits of the Study

The temporal development of biochemical parameters was well documented over a long period of therapy in most of the 20 new WD patients. The number of patients (n = 20) seems small; however, the primary selection criterion was “listed for liver transplantation”. Such patients are rare in a neurological department. They had been frequently monitored, which allowed us to analyze the improvements in their liver function during the first 4 months and their neurological outcomes over several years. The present study was retrospective and was performed on selected patients from a single specialized center. Therefore, a multi-center prospective study is recommended to confirm the differences in the recovery of the brain and liver under CET.

## Figures and Tables

**Figure 1 jcm-12-04861-f001:**
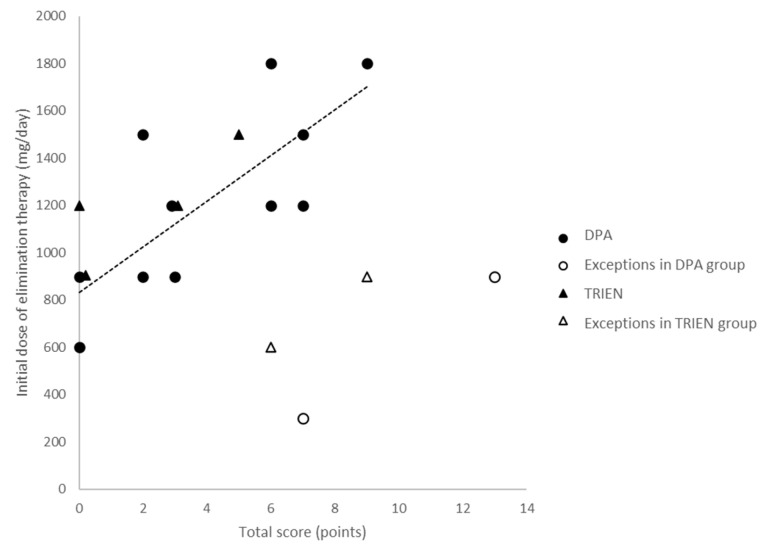
Relationship between the total score (TS; *x*-axis) and the dose of copper elimination therapy (DPA, circles; TRIEN, triangles; *y*-axis). Apart from two exceptions for DPA (open circles) and two exceptions for TRIEN (open triangles), there was a significant correlation between TS and the dose, which was significant (*p* < 0.01) only for DPA (full circles). The regression line was calculated for the relationship between TS and the dose of DPA.

**Figure 2 jcm-12-04861-f002:**
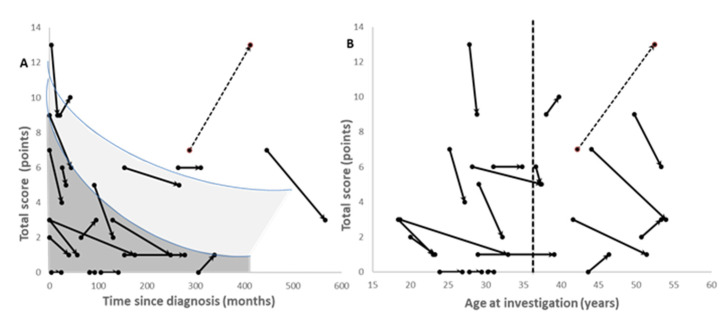
Temporal change in the total score (TS) depending on the time since diagnosis (**A**) and on age at investigation (**B**). The neurological symptoms responded well to therapy, and the total score was reduced to values below 3, indicating that the patients were only mildly affected (dark grey area in (**A**)). However, some patients remained moderately (2 < TS < 7) affected (light grey area in (**A**)). All patients below the age of 37 years (vertical line) improved (**B**). For patients above the age of 37 years (vertical hatched line), secondary worsening may occur (**B**). The hatched arrow indicates an exceptional case with a left-hemispheric stroke.

**Figure 3 jcm-12-04861-f003:**
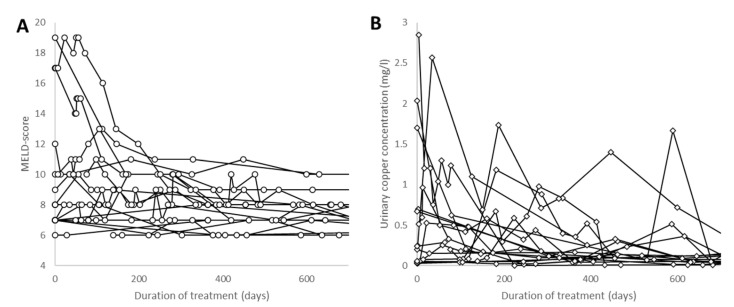
In all patients (except the patient who received a transplant), the model of end-stage liver disease (MELD) score declined to values below 12 after a treatment period of more than 200 days (**A**). The 24 h concentration of copper in the urine (**B**) showed high variability and declined to close to normal values (<0.03 mg/L) with increasing duration of therapy. The exceptionally high values are from patients with low compliance.

**Figure 4 jcm-12-04861-f004:**
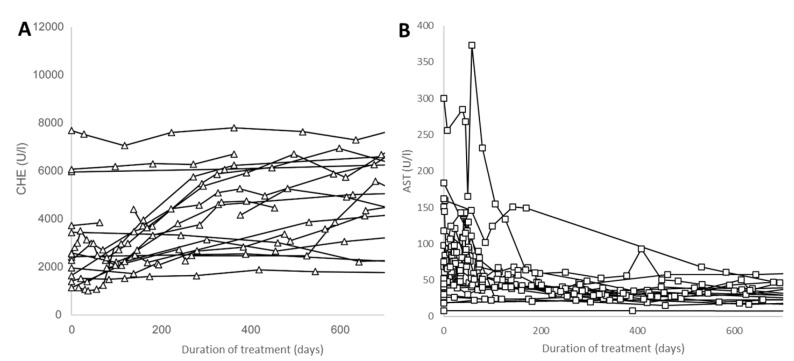
In most patients, pseudocholinesterase (CHE) continuously increased during therapy over a time period of up to 700 days (**A**). In one non-compliant patient, a further deterioration of CHE was observed despite treatment. In most patients, the liver enzyme aspartate transaminase (AST)-levels declined to close to normal values (around 50 U/L) within a treatment period of 200 days (**B**).

**Figure 5 jcm-12-04861-f005:**
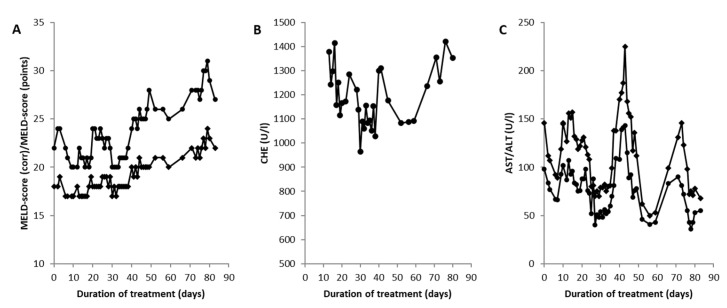
Temporal changes in the model of end-stage liver disease (MELD) score and the corrected MELD score (MELD score (corr)) (**A**) of serum levels of pseudocholinesterase (CHE) (**B**) and serum levels of aspartate transaminase (AST) and alanine transaminase (ALT) (**C**) in the patient receiving the transplant since the onset of copper elimination therapy (CET). CHE, AST, and ALT levels (**B**,**C**) started to improve 30 to 40 days after the onset of CET. The MELD score progressively worsened despite his liver function beginning to recover (**A**).

**Figure 6 jcm-12-04861-f006:**
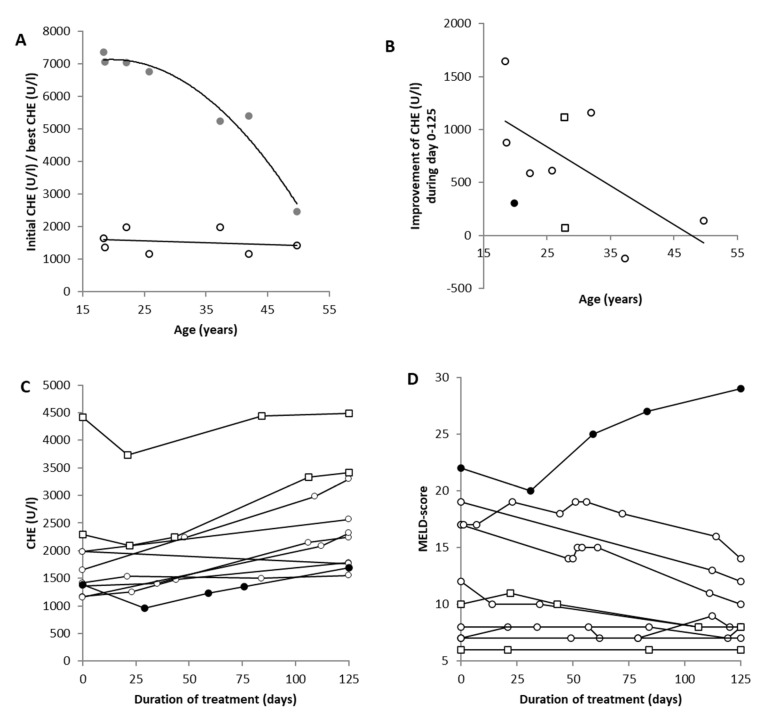
Comparison of the initial (open circles) and best (grey dots) serum levels of pseudocholinesterase (CHE) after the onset of copper elimination therapy (CET), revealing an age-dependent recovery of liver dysfunction (**A**). The recovery of CHE serum levels during the first 125 days of treatment was also age-dependent (**B**). Open circles in (**B**) indicate the values of the seven conservatively treated patients, the full circles indicate the data of the 19-year-old patient before transplantation, and the open squares are the values of the homozygotic twins. CHE significantly (*p* < 0.043) improved during the first 125 days of CET (**C**). With only one exception (the eldest patient), an improvement in CHE was observed in all patients, including the patient receiving the transplant (full circle) and the twins (open squares). Moreover, the model-end-stage-liver disease (MELD)scores of all patients improved or remained constant during the first 125 days of CET (**D**), except in the patient who had received the transplant (full circle in D) about 4 months after onset of CET.

**Table 1 jcm-12-04861-t001:** General clinical classification and results of the initial neurological investigation.

No. of Patients	Type	General Clinical Classification
1	Asymptomatic	No symptoms at all
3	Hepatic	1 patient with acute liver failure 2 patients with reduced daily activities and fatigue
1	Neuropsychiatric	Intellectual decline and moderate depression
15	Neurologic	7 patients with tremor of the extremities, head, and/or trunk 5 patients with Parkinsonian symptoms 3 patients with other movement disorders (cerebellar ataxia, chorea, generalized dystonia)
**No. of ** **Patients**	**Symptom**	**Initial Neurological Investigation**
		**Motor Symptoms**
14	Bradykinesia	Slowness in fast alternating movements of the fingers or tongue
9	Tremor	Clinically manifest tremors of the extremities, head, and/or trunk
8	Dysarthria	Dysarthria and/or dysphagia
7	Gait disorder	Spastic, cerebellar, or Parkinsonian gait disorder
7	Dystonia	Focal and/or generalized dystonia
6	Cerebellar	Ataxia of extremities
5	Oculomotor deficits	Oculomotor impairment or oculogyric crisis
		**Non-Motor Symptoms**
0	Sensory abnormalities	Sensory deficits of the legs, arms, hands, or fingers
3	Reflexes abnormalities	Reflex abnormalities (either enhanced reflexes with positive pyramidal tract signs or reduced or missing reflexes)
4	Neuropsychiatric symptoms	Neuropsychiatric symptoms; only 1 patient needed specific treatment

In column 2 the 7 motor subscores (Bradykinesia, Tremor, Dysarthria, Gait disorder, Dystonia, Cerebellar disorders, Oculomotor deficits) and the 3 non-motor subscores (Sensory abnormalities, Reflex abnormalities, Neuropsychiatric symptoms) of the Düsseldorf Wilson´s disease Score (DWDS) are listed.

**Table 2 jcm-12-04861-t002:** Improvement in the laboratory findings for the entire cohort and in two subgroups.

Parameter	Units	Initial Values		Final Values		Significance
		Initial MV	Initial SD	Final MV	Final SD	*p*<
24 h-PROT/L	mg/L	107.9	107.9	142.0	329.4	0.00049
24 h-PROT/d	mg/d	173.5	152.2	195.5	354.5	0.00604
CHE	U/L	3028	1886	5220	1379	0.00740
AP	U/L	131.0	71.6	87.3	25.7	0.01010
ALB (%)	%	54.5	6.6	60.2	2.7	0.02095
AST	U/L	78.8	76.2	31.6	17.3	0.04024
ALB (g/dL)	g/dL	6.1	7.6	4.4	1.0	0.04813
		**MELD** **≤ 10**	**Initial**	**MELD > 10**		**Significance**
**Parameter**	**Units**	**MV:** **Initial MELD** **≤ 10**	**SD:** **Initial MELD** **≤ 10**	**MV:** **Initial MELD > 10**	**SD:** **Initial MELD > 10**	***p*<**
Quick	%	91.1	5.93	54.5	16.07	0.00004
HB	g/L	14.73	0.69	12.66	1.14	0.00069
CHE	U/L	4502	1939	1849	567	0.00077
PTT	Sec	29.0	2.65	41.28	7.50	0.00106
ERY	10^6^/µL	4.97	0.31	4.14	0.54	0.00252
24 h-UCU/L	mg/L	0.16	0.20	1.31	0.98	0.00740
ALB (%)	%	58.9	3.56	50.5	6.2	0.00763
24 h-UCU/d	mg/d	0.26	0.36	2.41	1.89	0.00846
INR		1.04	0.05	1.55	0.44	0.00865
		**MELD** **≤ 10**	**Final**	**MELD > 10**		**Significance**
**Parameter**	**Units**	**MV:** **Final MELD** **≤ 10**	**SD:** **Final MELD** **≤ 10**	**MV: Final MELD > 10**	**SD:** **Final MELD > 10**	***p*<**
PTT	s	25.67	2.00	29.86	4.59	0.02172
CER	mg/dL	7.22	3.31	12.40	5.64	0.02910
TRANS	mg/dL	254.6	31.8	285.7	24.5	0.03343

MELD—model for end-stage liver disease; MV—mean value; SD—standard deviation; 24 h-PROT/L—concentration of protein in the 24 h urine collection; 24 h-PROT/d—urinary protein excretion over 24 h; CHE—pseudocholinesterase; AP—alkaline phosphatase; ALB (%)—percentage of albumin; AST—aspartate transaminase; ALB (g/dL)—albumin in g/dL; Quick—Quick´s test; HB—serum level of hemoglobin; ERY—erythrocyte count; 24 h-UCU/L—concentration of copper in the 24 h urine collection; 24 h-UCU/d—copper excreted in the urine over 24 h; INR—international normalized ratio; PTT—thromboplastin time; CER—ceruloplasmin; TRANS—transferrin.

## Data Availability

The datasets generated and/or analyzed during the current study are not publicly available because the datasets are part of a dissertation for a Doctor of Medicine but are available from the corresponding author upon reasonable request.

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
