# Peer review of "Differences in the Time Course of Recovery from Brain and Liver Dysfunction in Conventional Long-Term Treatment of Wilson Disease"

_jcm, 2023, doi:10.3390/jcm12144861_

Round 1

Reviewer 1 Report

In the manuscript entitled “Difference in the time course of recovery of brain and liver dysfunction in long-term treated Wilson disease”, the authors collected 20 WD patients who were considered for liver transplantation and observed for long time. The research came to the conclusions that for patients with acute liver failure, the first 4 months are critical and for patients with severe neurological symptoms, it is important that they are treated with fairly high doses over years.

The idea of this article is innovative, which provides the good suggestions for WD patients and doctors focused on this field. However, as an academic research, I think it has some defects in following details:

1.     The title of the article should be revised to highlight the changes of the patient's condition before and after liver transplantation.

2.     In the manuscript, the twins who were not transplanted should not be included, and if they are to be included, only the proband one should be retained.

3.     As I known, UWDRS (the Unified Wilson's Disease Rating Scale) is more widely used in WD patients and I recommended you to use this scale for future evaluations.

4.     The descriptive writing of basic information about the cohort in the manuscript (Results: 3.1) was complicated and redundant, which could be replaced by one or two tables.

5.     For the conclusions of this article, I think they are very meaningful. However, your citations are not comprehensive, there are a number of teams have published similar researches, like PMID: 32398357, PMID: 34997703, PMID: 27866189, etc.

6.     The language should be improved, and I suggest authors redraw the figures so as to be more normative and unified.

The language should be improved, and I suggest authors redraw the figures so as to be more normative and unified.

Author Response

In the manuscript entitled “Difference in the time course of recovery of brain and liver dysfunction in long-term treated Wilson disease”, the authors collected 20 WD patients who were considered for liver transplantation and observed for long time. The research came to the conclusions that for patients with acute liver failure, the first 4 months are critical and for patients with severe neurological symptoms, it is important that they are treated with fairly high doses over years.

The idea of this article is innovative, which provides the good suggestions for WD patients and doctors focused on this field. However, as academic research, I think it has some defects in following details:

1.      The title of the article should be revised to highlight the changes of the patient's condition before and after liver transplantation.

2.     In the manuscript, the twins who were not transplanted should not be included, and if they are to be included, only the proband one should be retained.

3.     As I known, UWDRS (the Unified Wilson's Disease Rating Scale) is more widely used in WD patients and I recommended you to use this scale for future evaluations.

4.     The descriptive writing of basic information about the cohort in the manuscript (Results: 3.1) was complicated and redundant, which could be replaced by one or two tables.

5.     For the conclusions of this article, I think they are very meaningful. However, your citations are not comprehensive, there are a number of teams have published similar researches, like PMID: 32398357, PMID: 34997703, PMID: 27866189, etc.

6.     The language should be improved, and I suggest authors redraw the figures so as to be more normative and unified.

Comments on the Quality of English Language

The language should be improved, and I suggest authors redraw the figures so as to be more normative and unified.

The authors are thankful to the reviewers for very careful reading and many helpful comments and suggestions to improve the manuscript.

Since we report on conventional treatment of WD-patients with hepatic und neurological symptoms we now highlight their conventional treatment in the title.

Only 1 patient was transplanted after 3 months of conventional treatment. We therefore think that all patients fit into the cohort.

One of the authors (HH) was involved in the development of the UWDRS. Therefore our team is familiar with the UWDRS. The UWDRS is suitable for scientific (cross-sectional) studies, but is too time-consuming for the use in long-term monitoring and clinical practice.

This section is shortened, and a table is added.

We have added these publications.

We have improved language and figures according to the suggestions of all 3 reviewers.

If further language problems exist, we will ask the editor for help with the language. One of the authors is a native speaker who also read the last version of the manuscript carefully.

Reviewer 2 Report

The idea of presentation consecutive newly diagnosed cases is interesting  to all who start working with WD and also to others having already experience.

The paper confirm in fact  the great variability of cases and good response to chelating therapy, which in most of cases safe patients from transplantation.

However to follow the paper in details is very difficult, afraid that almost impossible.

We  learn that patients had various neurological signs, but there is no simple  statement how many had  hepatic symptoms only, how many were asymptomatic, how many had neurology. It is not clear criteria for putting patients’ on waiting list for the transplantation.

Is not given how authors started therapy – since beginning full dose, or increase slowly, which dose was on entry, and which criteria were for dose modification. Fig 1 shows correlation of dose with severity – how dose was adjusted.  About it partially  one can learn from conclusions.

Discussion in fact contain a lot of results data. Some statements are bizarre eg. This protective function of the BBB leads to later manifestation of neurological symptoms compared to hepatic symptoms.

In fact only the part Conclusion part is clear.

Author Response

The idea of presentation consecutive newly diagnosed cases is interesting  to all who start working with WD and also to others having already experience.

The paper confirm in fact the great variability of cases and good response to chelating therapy, which in most of cases safe patients from transplantation.

However to follow the paper in details is very difficult, afraid that almost impossible.

We  learn that patients had various neurological signs, but there is no simple  statement how many had  hepatic symptoms only, how many were asymptomatic, how many had neurology. It is not clear criteria for putting patients’ on waiting list for the transplantation.

Is not given how authors started therapy – since beginning full dose, or increase slowly, which dose was on entry, and which criteria were for dose modification. Fig 1 shows correlation of dose with severity – how dose was adjusted.  About it partially  one can learn from conclusions.

Discussion in fact contain a lot of results data. Some statements are bizarre eg. This protective function of the BBB leads to later manifestation of neurological symptoms compared to hepatic symptoms.

In fact only th part Conclusion part is clear.

We agree.

We agree

Reviewer 2 is right in both aspects. A table is added providing the reader with more information on the symptoms of the patients.

All patients (except the twins) had been listed for LTX by several gastroenterologists. This has been mentioned. We used the “listing for LTX” as a inclusion criterium for recruitment, but did not check consistency of listing. 

Increase of dose during 3 weeks up to the full initial dose is mentioned now. Chelating agents were switched when antinuclear antibody increased or clinical side-effects occurred. This is mentioned now.

We avoid to mention results in the discussion which have not been presented in the result section.

To satisfy reviewer 2 the sentence about the protective function of the BBB is omitted now. But we don´t think it is bizarre, it was been mentioned in previous papers by other authors.

Reviewer 3 Report

Wilson's disease (WD) has long been considered a rare hereditary monogenic autosomal recessive disease with an unfavorable prognosis. However, the development of molecular genetics and the introduction of neonatal and population-based population screenings have allowed us to reconsider the view on this. On the one hand, in recent years it has been convincingly demonstrated that this disease is much more common than previously thought. On the other hand, improving the diagnosis of the disease stimulates the search for new approaches to therapy, primarily to the use of chelated compounds that contribute to the removal of excess copper from the patient's body. All this explains the relevance of the authors' research.

The article is designed in accordance with the requirements of the journal, it is well read, but technical correction is required, first of all, the style of the English language.

11)      Write the name of the gene in italics.

22)      Line 75: Please, instead of the phrase "In 1993 mutations of the ATP7B coding WD-gene on chromosome 13q14.3-q21.1», it is better to write «In 1993, causal mutations in the ATP7B gene (locus13q14.3-q21.1) responsible for the development of WD were identified».

33)      Avoid starting sentences with abbreviations.

44)      After the Introduction section, formulate a clear goal of your research.

55)      Lines 125-131, 157-158: Specify the units of measurement for assessing the symptoms of the disease according to the scales you use (for example, 3 points, 6 points, etc.).

66)      Please indicate which diagnostic equipment was used to perform biochemical analyses.

77)      Material and Methods section: Add the results of DNA testing. What causal gene mutations were found in the patients you observed? This may partially affect the severity of the disease, the rate of progression of the disease and the expected therapeutic response.

88)      Line 160: Please use the "p-value = 0.05" pattern here and further.

99)      Lines 169-171: Since the sample was small, and the variations in the age of patients and the age of manifestation of the disease were in a wide range, it is incorrect to use the average value of the age of patients. Please replace this parameter with median (Me) and interquartile interval (Q25;Q75).

110)   Line 172 and further: write "70%" instead of "=70%", and then follow this pattern.

111)   Figure 1, Figure 2: Write the name of the axes with a capital letter. Changing "mg" to "mg/day". Change "0-30" to "points".

112)   Figure 4: Write the name of the axes with a capital letter. Add the axis change unit - (days).

113)   Table 1: This needs to be corrected according to the table template. Divide the table into three separate tables OR combine all the parameters into one table (delete Tab. 1A, Tab. 1B, Tab. 1C).

114)   Under the Table 1, add a Note explaining all the abbreviations used in this table.

115)   Line 256, 305, etc.: You have already used the abbreviation WD before, so replace the full name of the disease with an abbreviation. In general, throughout the entire manuscript, authors should carefully review all abbreviations and use them after the first use.

116)   Figure 5: Write the name of the axes with a capital letter. Add the axis change unit - (points).

117)   Figure 6: Write the name of the axes with a capital letter.

118)   In the Discussion section, I recommend explaining why the authors chose this particular method of therapy and what are its advantages over other methods. How are the results of this study fundamentally different from previous studies?

119)   The authors cite references to studies that were published more than 10 years ago (22 of the 52 references are old, with the exception of references of historical interest). The author recommends revising the sections "Introduction" and "Discussion" taking into account the results of research in recent years (preferably no later than 5 years).

A moderate revision of punctuation and an extensive revision of the style of the English language is required

Author Response

Comments and Suggestions for Authors

Wilson's disease (WD) has long been considered a rare hereditary monogenic autosomal recessive disease with an unfavorable prognosis. However, the development of molecular genetics and the introduction of neonatal and population-based population screenings have allowed us to reconsider the view on this. On the one hand, in recent years it has been convincingly demonstrated that this disease is much more common than previously thought. On the other hand, improving the diagnosis of the disease stimulates the search for new approaches to therapy, primarily to the use of chelated compounds that contribute to the removal of excess copper from the patient's body. All this explains the relevance of the authors' research.

The article is designed in accordance with the requirements of the journal, it is well read, but technical correction is required, first of all, the style of the English language.

11)      Write the name of the gene in italics.

22)      Line 75: Please, instead of the phrase "In 1993 mutations of the ATP7B coding WD-gene on chromosome 13q14.3-q21.1», it is better to write «In 1993, causal mutations in the ATP7B gene (locus13q14.3-q21.1) responsible for the development of WD were identified».

33)      Avoid starting sentences with abbreviations.

44)      After the Introduction section, formulate a clear goal of your research.

55)      Lines 125-131, 157-158: Specify the units of measurement for assessing the symptoms of the disease according to the scales you use (for example, 3 points, 6 points, etc.).

66)      Please indicate which diagnostic equipment was used to perform biochemical analyses.

77)      Material and Methods section: Add the results of DNA testing. What causal gene mutations were found in the patients you observed? This may partially affect the severity of the disease, the rate of progression of the disease and the expected therapeutic response.

88)      Line 160: Please use the "p-value = 0.05" pattern here and further.

99)      Lines 169-171: Since the sample was small, and the variations in the age of patients and the age of manifestation of the disease were in a wide range, it is incorrect to use the average value of the age of patients. Please replace this parameter with median (Me) and interquartile interval (Q25;Q75).

110)   Line 172 and further: write "70%" instead of "=70%", and then follow this pattern.

111)   Figure 1, Figure 2: Write the name of the axes with a capital letter. Changing "mg" to "mg/day". Change "0-30" to "points".

112)   Figure 4: Write the name of the axes with a capital letter. Add the axis change unit - (days).

113)   Table 1: This needs to be corrected according to the table template. Divide the table into three separate tables OR combine all the parameters into one table (delete Tab. 1A, Tab. 1B, Tab. 1C).

114)   Under the Table 1, add a Note explaining all the abbreviations used in this table.

115)   Line 256, 305, etc.: You have already used the abbreviation WD before, so replace the full name of the disease with an abbreviation. In general, throughout the entire manuscript, authors should carefully review all abbreviations and use them after the first use.

116)   Figure 5: Write the name of the axes with a capital letter. Add the axis change unit - (points).

117)   Figure 6: Write the name of the axes with a capital letter.

118)   In the Discussion section, I recommend explaining why the authors chose this particular method of therapy and what are its advantages over other methods. How are the results of this study fundamentally different from previous studies?

119)   The authors cite references to studies that were published more than 10 years ago (22 of the 52 references are old, with the exception of references of historical interest). The author recommends revising the sections "Introduction" and "Discussion" taking into account the results of research in recent years (preferably no later than 5 years).

Comments on the Quality of English Language

A moderate revision of punctuation and an extensive revision of the style of the English language is required

We agree with this nice summary on WD.

Name of genes are written in italics now.

We are thankful for this help and follow reviewer´s advice.

That has been corrected.

We have rewritten the last paragraph of the Introduction.

Here and in the figures we add “points”.

This is added now.

The diagnostic procedures used to diagnose WD are now mentioned explicitly. Only a few patients underwent DNA testing.

So far no clear-cut genotype/phenotype correlation has been found in WD. Factors influencing the genotype have been summarized in the discussion.

The significance level is p=.05. This is mentioned in the Statistics-section. Higher levels of significance are mentioned explicitly.

As long as data of a cohort are normally distributed mean value and standard deviations are valid parameters. Since we have not performed a test on normal distribution, we follow reviewer 3´s advice to use non-parametric characteristics.

This is modified now.

This is modified now.

Thank you for very careful reading!

Table 1 is now Table 2 the subsections A,B,C are omitted now!

A note explaining the abbreviations is added.

We checked the use of abbreviations and found several problems. We therefore are very thankful for this suggestion.

This is modified.

This is modified.

During the initial phase of therapy we increase the dose within 3 weeks and use doses of 900mg/day or higher when a patient has neurological symptoms and is compliant. To be sure that this approach is superior to other approaches a comparative study has to be performed which is probably impossible to initiate.

The major results of the present study are the explicit estimations of durations of treatment necessary to achieve improvement of hepatic or neurological symptoms.

Several recent studies are mentioned in the manuscript now additionally.  

We think that the journal will help us to improve language problems. 

Round 2

Reviewer 1 Report

In this revised version, the author changed several language issues and clarified some contents more clearly. At the same time, the author gave answers to reviewers’questions. I have no additional comment on the content. But I strongly suggest that the author need the language polish. A few of medical terms are unprofessional, like de novo WD patients…

In this revised version, the author changed several language issues and clarified some contents more clearly. At the same time, the author gave answers to reviewers’questions. I have no additional comment on the content. But I strongly suggest that the author need the language polish. A few of medical terms are unprofessional, like de novo WD patients…

Author Response

In this revised version, the author changed several language issues and clarified some contents more clearly. At the same time, the author gave answers to reviewers’questions. I have no additional comment on the content. But I strongly suggest that the author need the language polish. A few of medical terms are unprofessional, like de novo WD patients…

In response to your suggestions, we have taken significant steps to enhance the quality of our manuscript. Firstly, we engaged a professional English editing service by MDPI to refine the language and structure. Following this, we also had a native English speaker review the document to ensure its readability and grammatical accuracy.

We believe these comprehensive revisions have greatly improved the manuscript.

Reviewer 3 Report

The authors have made changes to the manuscript and improved it. However, not all problems have been fixed. The purpose of the study is formulated vaguely and looks like a fragment of the "Discussion" section. This problem should be fixed by the authors.

Table 1 needs technical correction. The name of the first column is missing. I recommend writing all column and row names with a capital letter. Table 1 needs technical adjustment. The name of the first column is missing. I recommend writing the names of all columns and rows with a capital letter. The purpose of the text given under table 1 is not clear to me.

The reviewer's recommendations for improving the figures are not taken into account by the authors. The names of the axes are written with a small, then with a capital letter. There is still no explanation of the abbreviations used in Figures 3, 4, 5 and 6.

Table 2 needs technical correction. The note to table 2 needs technical correction. The note should be formatted as follows. Note: ABBREVIATION - full name; ABBREVIATION - full name; .....

Please remove abbreviations from the names of sections and subsections.

The number of outdated references is still very large, it does not correspond to the Guidelines for Authors. The authors automatically added new links to the previously cited old links without changing the content of the sentences in the text of the manuscript. This is unacceptable and discredits the publications of the authors of modern papers.

The manuscript needs a specialized revision of the English style.

The manuscript needs a special revision of the style of the English language.

Author Response

The authors have made changes to the manuscript and improved it. However, not all problems have been fixed. The purpose of the study is formulated vaguely and looks like a fragment of the "Discussion" section. This problem should be fixed by the authors.

Table 1 needs technical correction. The name of the first column is missing. I recommend writing all column and row names with a capital letter. Table 1 needs technical adjustment. The name of the first column is missing. I recommend writing the names of all columns and rows with a capital letter. The purpose of the text given under table 1 is not clear to me.

The reviewer's recommendations for improving the figures are not taken into account by the authors. The names of the axes are written with a small, then with a capital letter. There is still no explanation of the abbreviations used in Figures 3, 4, 5 and 6.

Table 2 needs technical correction. The note to table 2 needs technical correction. The note should be formatted as follows. Note: ABBREVIATION - full name; ABBREVIATION - full name; .....

Please remove abbreviations from the names of sections and subsections.

The number of outdated references is still very large, it does not correspond to the Guidelines for Authors. The authors automatically added new links to the previously cited old links without changing the content of the sentences in the text of the manuscript. This is unacceptable and discredits the publications of the authors of modern papers.

The manuscript needs a specialized revision of the English style.

The last paragraph of the introduction is modified again to present the aim of the study more clearly.

Reviewer 3 is right:
Table 1, its heading and the notes to Table 1 have been placed at a wrong position in the manuscript. This probably caused some irritations and is corrected now. We apologize for this.

We apologize for any confusion earlier. In the document version you reviewed, the original figures were retained in track mode, which might have led to some uncertainty. The previous versions of the figures have now been removed from the track mode, and the updated, enhanced figures have been incorporated into the current version of the document.

Reviewer 3 is also right in this aspect:
As is true for Table 1 also Table 2, its heading and its notes had been placed at a wrong position in the manuscript. This probably caused further irritations and is corrected now. We apologize for this. The notes were modified according to the suggestion of reviewer 3.

We removed all abbreviations from head lines.

The number of references is reduced to 50 again. Twelve references were removed. 10 of these references were published before 2014 and 6 before 2000. At least half of these publications are from our group. Thus, the number of “outdated” references is considerably reduced.

Because of the use of more recent publications further text had to be added to the manuscript.

In response to your suggestions, we have taken significant steps to enhance the quality of our manuscript. Firstly, we engaged a professional English editing service by MDPI to refine the language and structure. Following this, we also had a native English speaker review the document to ensure its readability and grammatical accuracy.

We believe these comprehensive revisions have greatly improved the manuscript.
